# Mechanism of Textural Reorganization in Silkworm Chrysalis and Pea Protein Extrusion: Structural Evolution and Quality Characteristic

**DOI:** 10.3390/foods14071134

**Published:** 2025-03-25

**Authors:** Xun Zhang, Min Wu, Tao He, Dongyu Sun, Huihuang Xu, Tianqi Zhang, Wenguang Wei

**Affiliations:** College of Engineering, China Agricultural University, P. O. Box 50, No. 17 Qinghua East Road, Haidian District, Beijing 100083, China; zhangxun@cau.edu.cn (X.Z.); hetao@cau.edu.cn (T.H.); sdymavis@163.com (D.S.); xhhhenan@163.com (H.X.); sy20233071641@cau.edu.cn (T.Z.); wwg5946@163.com (W.W.)

**Keywords:** extrusion, insect protein, meat substitute, alternative protein source

## Abstract

Mixed extrusion of animal and plant proteins has great potential in meat substitution studies. In this study, we analyzed the mechanism of change in the reorganization of animal and plant proteins during extrusion by exploring the changes in physicochemical properties with different percentages of silkworm chrysalis protein (SCP) additions (3%, 6%, 9%, 12%, 15%) mixed with pea protein isolate (PPI). The results showed that the moderate addition of SCP (12%) reduced the stiffness and denseness of the protein structure of the extrudates, and increased the total amino acid content of the extrudates, up to 74.83. Meanwhile, the addition of SCP changed the rearrangement of the proteins to form new chemical cross-linking bonds with higher bonding energies. Enthalpy of the sample up to 252.6 J/g, enhancing the denaturation energy requirement of the sample. Notably, the addition of SCP weakened the textural properties of the product, resulting in a minimum fibrous degree of 0.88, and improved the overall color of the sample, resulting in an L* value of up to 114.61. Such a change makes the product more suitable for further processing. Scanning electron microscopy (SEM) revealed that the addition of SCP changed the microstructure of the product, resulting in a looser, more porous sample overall. These results systematically elucidate the microscopic mechanisms of SCP and PPI restructuring during high-moisture extrusion.

## 1. Introduction

Animal proteins, such as meat, fish, and dairy, remain the primary source of human protein intake, driven by population growth and rising dietary protein needs. These proteins provide a balanced amino acid profile [1] and are easily digested and absorbed [2]. However, the livestock sector, which supports animal protein production, contributes 12% of global greenhouse gas (GHG) emissions [3], along with ecological and pollution challenges. In contrast, plant proteins, with their environmental benefits and varied sources, offer a sustainable pathway for protein development [4,5]. This has led to a surge of interest from researchers and consumers alike, making it a prominent area of focus in the scientific community. A substantial body of research has enhanced our comprehension of plant proteins [6,7]. To meet the very different requirements of the high amino acid content of animal proteins and the green, low-carbon safety of plant proteins, there has been a recent surge of interest in hybrid protein products combining both proteins, with a focus on finding suitable processing methods to develop innovative food products based on hybrid proteins combining both proteins.

High-moisture extrusion (HME) represents an emerging food-processing technology that converts protein materials into products with meat-like texture through the application of high temperatures and high pressure [8]. HME not only significantly enhances the taste and texture of proteins but also preserves their nutrient content, thereby meeting consumer demand for healthy, environmentally friendly, and sustainable foods. In recent years, a considerable advancement has been made in the study of HME of plant proteins, particularly in the processing of PPI [9,10,11]. A substantial body of research has demonstrated the efficacy of HME technology in facilitating the structural reorganization of plant proteins and the modification of protein products. The findings of Sun et al. [12] provide a compelling illustration of this phenomenon. The addition of konjac gum during HME of PPI [13] has been demonstrated to effectively enhance the texture characteristics of the extruded products, resulting in a gradual conversion of the protein extrusion flow pattern from plug flow to mixing flow.

The structural differences between animal and plant proteins are fundamental. They have disparate peptide sequences and native environments, as evidenced by Day et al. [14]. Consequently, their secondary and tertiary structures diverge, which underlies the disparities in their properties following processing. In general, animal proteins demonstrate superior structural flexibility and are more resistant to aggregation than plant proteins. The structural differences between animal and plant proteins mean that animal proteins still cannot be completely replaced by plant proteins. The addition of some animal proteins to the development of plant-based meats is the current option for the development of meat substitutes [15]. The flavor and nutritional value of the product are largely determined by animal proteins, while plant proteins are primarily responsible for their textural and structural properties [16]. Consequently, the interaction between animal and plant proteins during the production process is crucial for the success of the product. Among protein-processing technologies, extrusion has the advantages of high processing efficiency and low loss of nutritional value, which make extrusion a feasible method for modifying and restructuring mixed products of animal and plant proteins. However, further study is required to elucidate the microstructure and recombination mechanism of animal and plant proteins in the extrusion process. Recent studies by experts and scholars have investigated the mixed extrusion of animal and plant proteins, with a particular focus on fish [17], minced meat [18], and edible insects [19,20]. Goes et al. [17] mixed fish processing residue powder and maize flour for extrusion to create a snack food product. The results of their study demonstrated that the addition of fish processing residue powder had a beneficial impact on the protein content, micronutrient content, and nutritional value of the extruded snacks. Pöri et al. [18] successfully produced hybrid extrudates using a mixture of ground beef and PPI. These findings demonstrate that hybrid extrusion of animal and plant proteins is an effective and feasible approach to address the current limitations of plant protein products. Edible insects [21] have garnered attention as a source of animal protein due to their diversity, environmental compatibility, and rich nutritional profile. Insects possess a diverse range of protein, fat, and micronutrient content [22], and offer notable advantages over traditional animal husbandry practices in terms of greenhouse gas emissions and environmental pollution [23]. Focusing on mixed extrusion processing of insect proteins, some current results have been reported. Smetana et al. [19] mixed insect and soy proteins for high moisture extrusion, and the resulting SEM images demonstrated that the addition of insect proteins resulted in a more optimal texture for the extruded product. The study by Segovia et al. [20] indicated that the use of insect and PPI powders as alternative protein and mineral sources represents a promising and innovative avenue for further investigation. Nevertheless, the extant studies on the hybrid processing of animal and plant proteins remain concentrated on the examination of the extrusion process and the outcomes of textural and structural enhancement. However, there is a paucity of scientific conclusions and mechanistic explanations pertaining to the microstructural alterations of animal and plant meat proteins during extrusion and the conformational evolution of the two proteins reconstituted under elevated temperatures and pressures.

Nowadays, edible insects are recognized as a new source of protein, and silkworm pupae are one of the edible insects. The silkworm pupa is a biological and economic insect of the order Lepidoptera. In China, silkworm pupae are a popular food among consumers due to their high protein content [24] and rich amino acid profile. Silkworm pupae are considered a high-quality source of animal protein and are the only insect food on the list of novel food resources issued by the Chinese Ministry of Health [25], silkworm pupae have also been used as food and industrial raw materials due to their high nutritional value and diverse bioactivities [26]. Despite the many advantages of silkworm pupae, their palatability and sensory acceptability as human food still needs to be further improved [26], hence the concept of “invisible insects” has been proposed, and the incorporation of insects into the food preparation process could be a solution to these difficulties. Positive results have been obtained from the use of silkworm pupa proteins in the production of batters and protein-rich biscuits [27,28]. HME is also one of the technologies that could be used to realize this concept.

Hybrid protein extrusion may prove an effective solution to the challenges associated with plant protein extruded products. The combination of PPI and SCP in HME is anticipated to enhance the texture and nutritional value of the product through protein complementary effects. It is therefore essential to investigate how SCP and PPI interact under high-temperature, high-shear extrusion conditions, hypothesizing that their combined structure may reorganize in ways that enhance the overall quality and functional potential of the extruded products.

This study examined the conformational changes in SCP and PPI under high-moisture extrusion at elevated temperatures and pressures. The mixed extruded products were subjected to a comprehensive evaluation, employing a range of analytical techniques, including texture analysis, FTIR analysis, thermal property analysis, amino acid type analysis, rheological analysis, color analysis, and microscopic and macroscopic structure characterization. The evolution of the two protein recombination mechanisms during the extrusion process of PPI and SCP was elucidated in great detail, thereby providing a novel theoretical reference and practical basis for the determination of the optimal mixing ratio of SCP and PPI proteins.

## 2. Materials and Methods

### 2.1. Materials

PPI was supplied by Yantai Shuangta Food Co., Ltd. (Yantai, China) with a protein concentration of 82.01 ± 0.54%. SCP was procured from Ningnuo Trading Co., Ltd. (Shijiazhuang, China) with a protein concentration of 91.05 ± 0.23%. All the standards, chemicals, and reagents utilized in this study were obtained from McLean Biochemical Co., Ltd. (Shijiazhuang, China).

### 2.2. Preparation and Formulation of Raw Materials

The protein blends employed in the study comprised SCP and PPI, which were blended for 20 min using a blender (Kele Machinery Equipment Co., Ltd., Zhengzhou, China). Samples of varying concentrations were prepared. SCP and PPI were mixed at different mass ratios (*w*/*w*), and after a number of preliminary pre-tests, it was finalized that SCP accounted for 0%, 3%, 6%, 9%, 12%, and 15% of the total mixture, respectively. The PPI was utilized as the control sample (CK).

### 2.3. High-Moisture Extrusion Process

We conducted the extrusion process using a twin-screw extruder (TwinLab-F 20/40, Brabender, Oberhausen, Germany), with the hopper and water inlet situated at a distance of 3 cm and 24 cm from the commencement of the screw, respectively. The screw L/D ratio was 40:1, the screw diameter was 20 mm, the screw speed was set to 100 rpm for mixing the powder, the feed speed was set to 34 rpm, and the water content was set to 60%. The temperatures from the mixing zone to the front melt zone are maintained at 50 °C, 70 °C, and 100 °C, respectively. The temperature of the back melt zone and the die mouth zone are identical and are referred to as the barrel temperature, which is set at 130 °C. The end of the extruder is equipped with a 300 mm cooling die (cooling zone) with a runner cross-section size of 25 × 7 mm, and the temperature is maintained at 70 °C by circulating water. When the extruder was operated under steady-state conditions, the extrudates were collected, and some of the fresh samples were vacuum-packed and stored in a refrigerator set at 4 °C. The remaining samples were freeze-dried using a vacuum freeze-dryer (ST85B3, Millrock Technology, Inc., Kingston, NY, USA). The freeze-dried samples were pulverized and passed through an 80-mesh sieve prior to storage in sealed containers.

### 2.4. Water-Holding Capacity (WHC)

The extruded mixed protein powder (0.2 g) was dispersed in 5 mL deionized water, mixed using a vortex mixer (VORTEX-5, Killing Bell Instruments, Inc., Haimen, China), and stored at 4 °C for 24 h. Following centrifugation at 3000× *g* for 15 min, the supernatant was removed, and the weight of the centrifuge tube and the residue was recorded. The WHC was calculated according to Equation (1):(1)$$WHC=\frac{M_{w2}-M_{w1}}{M_{w0}}$$
where *M_w_*_2_ is the weight of the tube and remainder (g), *M_w_*_1_ is the weight of the tube and dry protein sample (g), and *M_w_*_0_ is the weight of the dry protein sample.

### 2.5. Fourier Transform Infrared Spectroscopy (FTIR)

The Fourier transform infrared (FTIR) spectra of extruded mixed protein samples were determined utilizing a Spectrum 100 spectrometer (PerkinElmer, Norwalk, CT, USA). The extruded protein powder (2 mg) was combined with 0.2 g KBr, ground, and compressed into transparent sheets. Each sample was scanned in the range 400 cm^−1^ to 4000 cm^−1^ at a resolution of 4 cm^−1^. Baseline correction and normalization of the spectra were performed using the Omnic 8.2 software (Thermo Electric Corporation, Chicago, IL, USA). Fourier deconvolution function analysis was conducted with the Peak Fit v4.12 software (Origin Lab Corp., Waltham, MA, USA) to determine the relative content of the secondary structure of the extrusion protein amide I region (1700–1600 cm^−1^).

### 2.6. Color Testing

The color parameters (*L**, *a**, *b**) of the protein blend extrudates were evaluated utilizing a spectrophotometer (Hunterlab, Labscan XE, Reston, VA, USA), which was calibrated with a black calibration plate and a white calibration plate, respectively, prior to measurement.

### 2.7. Texture Properties

The textural properties of the mixed protein extrudates were assessed using a TA.XTplus texture analyzer (Stable Micro System Ltd., Godalming, Surrey, UK). The extrudates were modified in accordance with the methodology proposed by Chen et al. [29] The sample was cut into a square measuring 10 × 10 mm with a thickness of 7 mm. In TPA mode, the P/36R probe was utilized to compress twice, thereby simulating the oral chewing motions of a human being, and the strain was pressed to 50% at a test speed of 1.0 mm/s. The hardness, springiness, and chewiness of PPI extrudates were determined. Here, each sample was measured nine times to maintain the accuracy of the texture data. In order to perform the fibrous degree test, the HDP/BS probe was employed to cut the extrudates in both vertical and horizontal directions at a loading rate of 1 mm/s at 75% strain, with the maximum shear values subsequently recorded. Six tests were required for each sample to ensure accurate data. The fibrous degree was then calculated by dividing the transverse shear force by the longitudinal shear force [30].

### 2.8. Scanning Electron Microscope (SEM)

The extruded mixed protein powder was attached to a plate with double-sided conductive tape for one minute using gold spray (JEOL, JFC-1600 Auto Fine Coater, Tokyo, Japan). The surface morphology was observed using scanning electron microscopy (SEM) (SU3500, Hitachi, Tokyo, Japan) at an accelerating voltage of 15 kV and a magnification of 1000 times.

### 2.9. Rheological Properties

Rheological tests were conducted utilizing a TA ARES-G2 rheometer (Discovery HR-20, TA Instruments Co., New Castle, DE, USA). A mixed protein solution with a concentration of 25% was configured for the test. The rheological test employed a 50 mm APS Peltier (stainless steel HB). Following gap calibration at 25 °C, the measurement gap was set to 1 mm. Prior to the test, a circle of silicone oil was applied around the Peltier to prevent the sample from evaporating.

#### 2.9.1. Construction of the BP Neural Network

Steady-state shear tests were conducted at room temperature (25 °C). The shear rate range was 1–100 s^−1^, and the power law model was employed to fit the relationship between shear rate *γ* (s^−1^) and shear force *τ* (Pa) [31]. The relationship between the two is shown in Equation (2):(2)$$\tau =K\cdot \gamma^{n}$$
where *τ* is the shear stress, *K* is the consistency index, *n* is the flow behavior index, and *γ* is the shear rate.

#### 2.9.2. Small Amplitude Oscillation Shear Stress Measurement (SAOS)

The frequency sweep test is conducted within the angular frequency range of 1 rad/s to 100 rad/s. A strain of 0.2% is selected (less than 1% within the linear viscoelastic region). The curves of the storage modulus *G*′, loss modulus *G*″, and loss factor *tanδ* of the protein solution with frequency were recorded. The loss factor *tanδ* represents the ratio of the viscous part to the elastic part, calculated as follows (Equation (3)) [32]:(3)$$\tan\delta =\frac{G^{\prime \prime }}{G^{\prime }}$$
where *G*′ denotes the energy storage modulus of the protein solution, and *G*″ denotes the loss modulus of the protein solution.

### 2.10. Thermal Characteristics Test

The thermal properties (denaturation temperature *T_d_* and denaturation enthalpy Δ*H*) of protein extrudates were determined by differential scanning calorimetry (DSC Q20, TA Instrument, New Castle, DE, USA). The extruded 3–8 mg mixed protein powder was accurately weighed and sealed in an aluminum crucible. The empty crucible was used as a reference and heated from 20 °C to 120 °C at a rate of 10 °C/min.

### 2.11. Amino Acid Test

The amino acid content of the extrudate was determined using an amino acid analyzer (L-8900, Hitachi, Japan) in accordance with the methodology proposed by Brishti et al. [33]. The extruded mixed protein powder (0.1 g) was dissolved in 10 mL hydrochloric acid (6 mol/L) and four drops of phenol and sealed in a reaction tube with nitrogen for acid hydrolysis. The solution is heated in a furnace at 110 °C for 24 h. Following filtration, the solution is completely dried with an evaporator. Subsequently, 1.0 mL of sodium citrate buffer (pH 2.2) is added and filtered with a 0.2 μm membrane.

### 2.12. Data Analysis

All experiments were conducted in triplicate, and the resulting data were expressed as mean ± standard deviation unless otherwise stated in the method. The statistical analysis was performed using the SPSS 20.0 software package (SPSS Inc., Chicago, IL, USA). One-way analysis of variance (ANOVA) was employed, and the Duncan test was used to identify significant differences between the samples (*p* < 0.05).

## 3. Results and Discussion

### 3.1. Apparent Characteristics and SEM Analysis

The microstructure of the extrudates can be observed through the SEM. The results of the SEM are presented in Figure 1. The unprocessed PPI and SCP particles exhibited a spherical structure of varying sizes, which were irregularly distributed in the natural state. The extrusion process significantly altered their microstructure, resulting in the spherical protein powders gradually denaturing into flake-like morphology with a porous structure. It can be observed that the extrudates retain a recognizable porous and fine texture when SCP is not added. However, with the increase in SCP addition, the tiny dense pores gradually transform into larger pores, exhibiting the maximum porous structure. The addition of SCP at a level of 15% is consistent with the findings of Smetana et al. [19], which indicate that the unfolding and aggregation of protein chains during extrusion directly affects the microstructure of the product. PPI was observed to form a compact fibrous structure during extrusion, thereby maintaining a fine porous texture. However, the addition of SCP resulted in the disruption of this fibrous network, leading to weakened interactions between the protein chains and the formation of a looser structure. This, in turn, resulted in the formation of larger pores, a finding that is consistent with the study of the WHC of the samples.

Figure 2 demonstrates that the incorporation of SCP has a significant impact on the profile structure of the extruded products. It can be observed that the samples exhibit a distinct anisotropic structure in the absence of SCP, whereas the addition of SCP results in a reduction in this anisotropic structure. Furthermore, the anisotropic structure is related to the fibrous degree of the product, a phenomenon that is often accompanied by a decrease in the fibrous degree of the samples. This is in line with the conclusions of the analysis of textural properties.

### 3.2. Color Analysis

As a significant sensory attribute, color exerts a considerable influence on consumer acceptance and can indirectly reflect the degree of chemical reaction [34]. As evidenced in Table 1, the incorporation of SCP resulted in a notable enhancement in the brightness value (*L**) of the extrudate, accompanied by a reduction in the *a** and *b** values. This may be attributed to the chromatic discrepancy between SCP and PPI. SCP is characterized by a white hue, whereas PPI is typically observed to be pale yellow. Consequently, the incorporation of SCP results in an enhancement of the mixture’s chromatic brightness, leading to a notable elevation in the brightness value (*L**). Furthermore, the Maillard reaction represents the primary chemical process responsible for product browning [35]. The degree of this reaction is influenced by specific amino acids, such as lysine, which are present at lower levels in SCP. Consequently, the incorporation of SCP may result in a reduction in the extent of browning. The degree of Maillard reaction during extrusion affects the color of the extrudate, as indicated by lower *b** and *a**, which suggests that the addition of SCP weakens the color of the extrudate. The market favors light-colored extrudates, as they are more straightforward to color, facilitate further processing, and are more readily accepted by consumers.

### 3.3. Texture Analysis

Texture characteristics represent a significant physical index for the simulation of taste properties in extruded products. Table 2 illustrates the impact of varying SCP additions on the textural characteristics of extruded products. It can be observed that the incorporation of SCP results in a reduction in the hardness, resilience, and chewiness of extruded products. This is attributed to the relatively complex nature of SCP molecules and the lengthier protein chain segments [36]. The addition of SCP promotes the expansion of the mixed protein chain during extrusion. However, the mixed protein chain is unable to form a robust protein gel network through cross-linking in the die mouth and molding areas. In comparison with the ordered fibrous structure formed by pure PPI extrusion products, the protein gel network strength of the mixed extrusions was found to decrease following the addition of SCP. The protein gel network plays an important role in determining structural hardness and resilience [37]. Therefore, the weakening of this network will result in a softer texture, reduced resilience, and chewiness. Extruded samples with the addition of SCP were softer in texture compared to the pea protein extrudates produced in the existing study [13]. In the study by Wang et al. [38], the impact of incorporating silkworm pupa peptide on yogurt fermentation and quality was examined. The findings indicated that the addition of silkworm pupa peptide in quantities greater than 0.7% resulted in a reduction in the texture characteristics of yogurt. In a separate study, Segovia et al. [20] utilized extrusion technology to produce snacks by mixing corn grits with alphitobius diapering. The results demonstrated that the incorporation of alphitobius diaperinus led to a deterioration in the textural properties of the snacks. These results are also analogous to the findings of our study.

Figure 3 illustrates the impact of SCP supplementation on the fibrous degree of the extrudate. Cutting against the grain in animal meat has been shown to require higher cutting forces [39]. Furthermore, a fibrous degree greater than 1 indicates that cutting the extruded sample necessitates higher transverse cutting forces, suggesting that the fibrous structure is formed along the longitudinal direction, a phenomenon analogous to that observed in real meat [40]. Lee et al. [39] utilized a mixture of PPI and soy protein isolate, and HME was performed, resulting in a fibrous degree of the extruded products ranging from 0.9 to 1.2, which is analogous to our fibrous degree distribution. It can be observed that the incorporation of SCP resulted in a reduction in the fibrous degree of the extruded sample, from 1.15 to 0.89; the longitudinal shear force and transverse shear force of the extruded sample exhibited a tendency to decrease with the continuous addition of SCP. This can be attributed to the fact that during the extrusion process, SCP exhibits a poor aggregation effect in the molding zone due to its inherent amino acid composition and secondary structure, as well as the low content of sulfur-containing amino acids. The content of sulfur-containing amino acids in SCP is low, and the blended proteins after the addition of SCP are unable to form a high-strength protein structure through disulphide bonding. Consequently, an increase in the ratio of SCP is not conducive to the formation of a reticulated fibrous structure with high stiffness in the molding zone, nor to the densification of the blended proteins. This ultimately results in a reduction in the fibrillation of the extrudate, which is in accordance with the findings of Sun et al. [41].

### 3.4. FTIR Analysis

The infrared spectral features can be used to reflect changes in the functional groups of the samples before and after extrusion, which is an important factor in the development of the processing mechanism. Figure 4 illustrates the infrared characteristics of the samples following extrusion with varying ratios of SCP. The spectrograms revealed a consistent trend across all samples with varying addition ratios, indicating that no new chemical bonds were formed. This suggests a similarity in the chemical structures of the samples, which aligns with the findings of previous studies by Gao et al. [42] and Smriti et al. [43].

A pronounced absorption peak resulting from O-H expansion vibration was observed in all extrudates within the range of 3500–3300 cm^−1^. This suggests the presence of hydrogen-bonding interactions in the mixed extrudate proteins [44,45]. The peaks of the extrudates at different SCP additions were observed to be in the range of 3287.0–3297.0 cm^−1^. Furthermore, the peaks of the samples with the addition of SCP were found to be significantly red-shifted when compared to the PPI extrudate. The highest peak was reached at the addition of SCP of 12% at 3297.0 cm^−1^. This suggests that the addition of SCP weakened the extrudates. The addition of SCP resulted in a weakening of the stability of hydrogen bonding in the extruded product. This was attributed to the fact that the hydrophobic residues of SCP may have interfered with the formation of hydrogen bonding between the water-soluble fragments of PPI, thereby reducing the hydrogen-bonding force between the protein molecules. The introduction of trace amounts of SCP may have resulted in the disruption of hydrogen-bonding interactions between PPI molecules, thereby reducing the stability of the protein network. This was followed by a blue shift in the absorption peak, which is likely attributable to the aggregation of SCP molecules, leading to the formation of a separate structural network or even the reorganization and formation of new stable hydrogen bonds with the addition of more SCP.

By comparing the effect of the change in the amide I region of the infrared spectra of different SCP additions, it is possible to observe the secondary structure changes in the samples. Table 3 presents the influence of different SCP dosages on the secondary structure of the extrudate. The β-sheets secondary structure is the most prevalent, comprising approximately 60% of the total, followed by the α-helix, which accounts for approximately 23%. It can be observed that the α-helix and β-turns of the extruded samples exhibited a slight decrease at the initial stage of the addition of SCP, while the degree of β-sheets and random coil demonstrated a slight increase. This is attributed to the formation of a more disordered spatial conformation at the initial stage of the addition of SCP, which is a consequence of the intermolecular polymerization of different kinds of protein molecules during the extrusion process. As the quantity of SCP incorporated increased, the β-sheets and random coil structures of the extruded samples underwent a gradual transition to α-helix and β-turns. This suggests that as the amount of SCP added increased, the β-sheets ratio of the protein molecular chain decreased and the α-helix ratio increased. The ratio of α-helix to β-sheets increased to 0.4015, indicating that the addition of SCP would result in a more extensive stretching of the mixed protein molecular chain and enhanced flexibility of the mixed protein molecular chain. Upon reaching a concentration of 15%, the random coil structure of the sample increased, while the ratio of α-helix to β-sheets decreased. This suggests that an excess of SCP would impair the flexibility of the mixed protein molecular chain. The addition of SCP reduced the overall flexibility of the molecular chains of the protein blend compared to the PPI extrudates, a property that gradually reverted with increasing SCP content in the extrudates, returning to the closest match to the PPI extrudates at 12%, which was disrupted by the addition of excessive amounts of SCP.

### 3.5. Rheological Analysis

#### 3.5.1. Steady Shear Test Analysis

A further understanding of the structural reorganization and processing mechanism can be gained through the analysis of the fluidity of the sample fluids. Figure 5 illustrates the impact of varying ratios of SCP addition on the apparent viscosity of the extrudates. Furthermore, in order to more accurately reflect the trend of the samples, Table 4 demonstrates that the static rheological properties of the extrudates with varying SCP addition ratios were modeled using a power law. The fitting results indicate that the consistency coefficient (*K*) of the samples exhibited a trend of initial decrease and subsequent increase, reaching a minimum at 12% of the SCP addition, with a minimum value of 3.33 Pa·s^n^. The flow behavior indices (*n*) of all the feedstocks were less than 1, indicating that the sample solutions were all pseudoplastic fluids [46]. These results are consistent with previously reported flow curves for pea and soy proteins [47]. The data presented in Figure 5 demonstrate that the apparent viscosity of all groups of feedstocks exhibited a declining trend with increasing shear rate, as substantiated by the shear dilution behavior [48]. It is noteworthy that the apparent viscosity of the mixed extrudates also demonstrated a declining trend with increasing SCP addition, reaching its lowest point at 12% SCP. This indicates that the incorporation of SCP facilitates the polymerization of diverse protein molecules, resulting in a gradual transition from protein aggregation to stretching [49]. As the proportion of SCP increased, the apparent viscosity increased when the proportion of SCP reached 15%. This was due to the fact that the SCP molecules began to aggregate during the extrusion process after the addition of excessive SCP. This resulted in the protein chain as a whole showing a dense and aggregated state once more. In conclusion, the addition of SCP at different ratios reflects a trend whereby the protein chains of the samples undergo a series of changes, from aggregation to stretching and then aggregation. The changes in the state of the protein chains demonstrated by the static rheology are consistent with the conclusions of the FTIR analyses presented in the previous paper.

#### 3.5.2. Small Amplitude Oscillation Shear Stress Measurement Analysis

The results of the frequency scanning tests demonstrated the variability in rheological properties exhibited by multiple groups of samples when subjected to disparate shear conditions. Figure 6 illustrates the variations in sample storage modulus (*G*’), loss modulus (*G*″), and loss tangent *tanδ* with frequency. It can be observed that both the elastic modulus *G*′ and loss modulus *G*″ of all extruded products increase with increasing angular frequency (ω). This phenomenon was attributed to transient disaggregation of the molecular chains undergoing a short period of oscillation during the scanning frequency range [50], and *G*′ is consistently larger than *G*″, indicating that all sample solutions exhibit greater elasticity [51]. Additionally, *tanδ* < 1 demonstrates this behavior, and all samples display gel-like characteristics.

The incorporation of SCP resulted in a reduction in *G*′ and *G*′′ values, indicating that the addition of SCP led to an overall increase in the looseness of the extrudate and a reduction in cross-linking. This process results in a reduction in the storage of energy within the extrudate, as well as a reduction in the dissipation of energy. Consequently, the energy storage modulus and loss modulus of the extrudate decrease. Concurrently, the incorporation of SCP resulted in an initial increase in the loss angle *tanδ*, followed by a subsequent decline. This observation suggests that the introduction of an appropriate SCP can potentially enhance the fluidity of the samples. One plausible explanation for this phenomenon is that the addition of SCP at a low concentration may facilitate the unfolding of the protein molecular chains present in the extrudates. This unfolded chain structure has been demonstrated to reduce the density and rigidity of the protein network. This result also corroborates the conclusion drawn from the static rheology of the samples. The maximum value of *tanδ* is reached at 12% SCP addition, after which *tanδ* decreases. This indicates that the addition of excess SCP inhibits the fluidity of the extrudate, because the hydrophobic proteins gradually increase with the increase in excess SCP. This change reduces the diffusivity and solubility of the extrudate in water, thus hindering the viscoelastic change in the solution.

### 3.6. Water-Holding-Capacity Analysis

WHC is a fundamental attribute of the samples, which is contingent upon the intrinsic nature of the extrudates. Figure 7 illustrates the impact of varying SCP additions on the WHC of extrudates. It can be observed that as the proportion of SCP additions increases, there is a slight decline in the WHC of the samples. This phenomenon was attributed to the superior water-binding capacity of PPI, which enables it to form a more stable structure with water. However, the poor water solubility of SCP resulted in its inability to effectively bind water during extrusion, leading to a decrease in the water-holding capacity. From the perspective of protein structure, the reduction in WHC can also be attributed to alterations in the network structure of extrudate proteins. With the increase in SCP addition, PPI molecules and SCP molecules lack enough disulfide bonds for cross-linking reactions, and the hydrophobic residues of SCP molecules also affect the formation of hydrogen bonds in the rearrangement process of SCP molecules and PPI molecules, which results in the strength of the network structure of extrudate proteins diminishing, and the capacity to capture water molecules reducing, which ultimately results in a decline in WHC. This is consistent with the changes observed in the textural analyses, and similar findings were reported in the study by Zhang et al. [52].

### 3.7. Amino Acid Analysis

Table 5 illustrates the amino acid profiles of the samples with varying SCP additions. The six most prevalent amino acids in the extrudates of all fractions were glutamic acid, aspartic acid, leucine, arginine, lysine, and phenylalanine. It is evident that the total amount of amino acids in the extrudates increased with the addition of SCP. This is due to the fact that SCP has a richer amino acid content compared to PPI. The hydrophilic and hydrophobic amino acid content of the samples increased with the addition of SCP, which can also be attributed to the higher content of hydrophilic and hydrophobic amino acids in SCP itself. It is noteworthy that the proportion of proline, glycine, and alanine exhibited a marked increase in the samples with SCP in comparison to those without SCP. Furthermore, the rise in glycine and alanine levels had a favorable impact on the stability of the protein structure [53]. The proline side chain is capable of forming a ring structure, and an increase in proline has a beneficial effect on the stability of the structure and folding form of the sample [54]. In conclusion, the results of the amino acid analysis demonstrated that the blending of SCP and PPI could enhance the overall content of amino acids.

### 3.8. Thermal Analysis

The application of differential scanning calorimetry allows for the monitoring of structural alterations in proteins resulting from processing. Furthermore, it can be employed as a fundamental methodology for the detection of heat-induced protein denaturation or protein structure unfolding. The denaturation temperature (*T_d_*) is indicative of the thermal stability of the protein, whereas Δ*H* is indicative of the energy requirement of the protein for denaturation. Table 6 illustrates the alterations in the thermal characteristics of the samples comprising different components. It can be observed that the incorporation of SCP resulted in a decline in the thermal stability of the product, while concurrently, a notable elevation in the enthalpy of the samples was evident (*p* < 0.05). Consequently, the enthalpy of the samples increased from the highest recorded value of 137.30 J/g to a new maximum of 252.60 J/g. This indicates that the mixed protein molecules may form new chemical cross-linking bonds between the protein molecules with higher bonding energy [55], which would require more energy to break [56]. The highest Δ*H* was reached at 12% SCP addition. The enthalpy decreased at the 15% SCP addition, yet remained higher than that of the pure PPI extrudate. In conclusion, the DSC data demonstrated that the incorporation of SCP altered the protein conformation, necessitating elevated thermomolecular mechanical energy for destruction, while concurrently forming novel chemical cross-linking bonds with augmented bonding energies. This resulted in a heightened energetic demand for denaturation, reflecting a substantial alteration in the conformation of the sample proteins following SCP addition.

## 4. Conclusions

This study demonstrates that modifying the SCP ratio in PPI under high-moisture extrusion significantly impacts the structural and functional characteristics of the extrudates. The incorporation of moderate SCP levels (12%) optimally reorganized the protein matrix, facilitating new cross-linking bonds with increased bonding energy, as evidenced by a peak enthalpy of 252.60 J/g. This restructuring also improved the amino acid composition, elevating proline, glycine, and alanine levels. Furthermore, the incorporation of SCP modulated the rheological properties, resulting in a decrease in density, stiffness, and apparent viscosity at a 12% addition, which was followed by recovery. This reflected the dynamic restructuring of the protein network. Microstructural analyses demonstrated a transformation in pore size and distribution, creating larger, more porous textures at higher SCP levels (up to 15%), which reflects a tendency for the extruded product structure to be looser, reducing the water-holding capacity of the extruded product. These structural modifications enhanced the color, making them more adaptable for further processing. This study provides insights into the structural reorganization mechanisms of SCP-PPI mixtures. Further research should be conducted to optimize extrusion parameters and investigate bioavailability changes in SCP and PPI, with the aim of developing sustainable protein substitutes.

## Figures and Tables

**Figure 1 foods-14-01134-f001:**
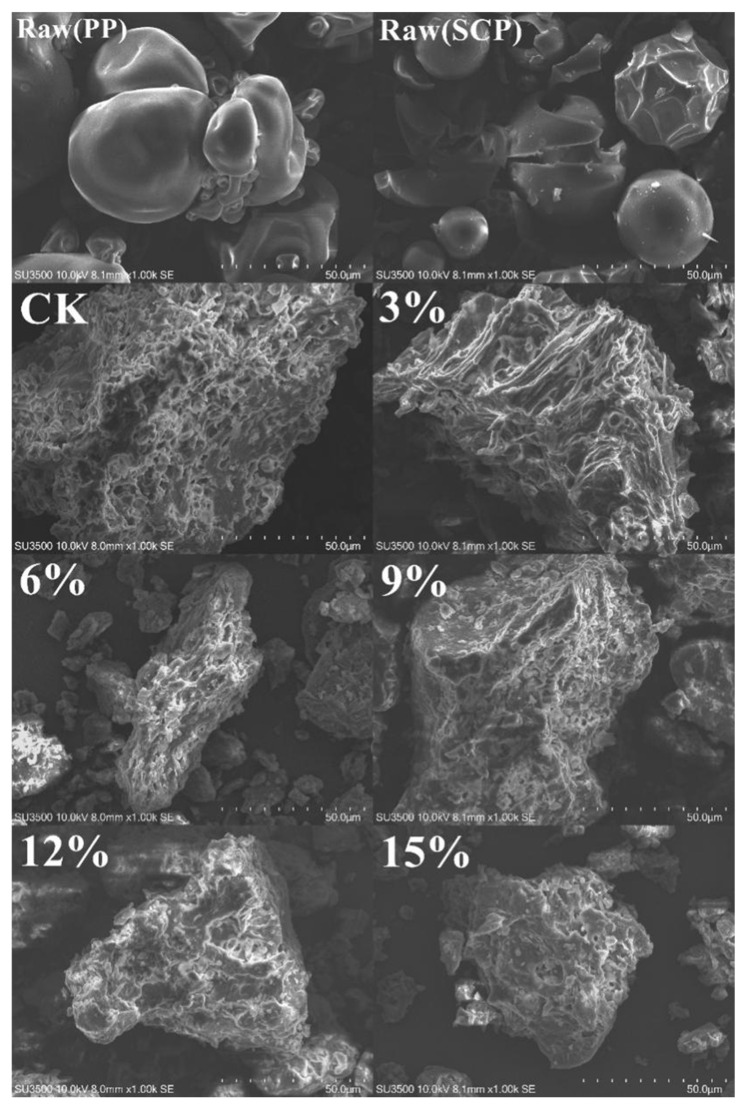
Microstructure of extrudates with different percentages of SCP addition.

**Figure 2 foods-14-01134-f002:**
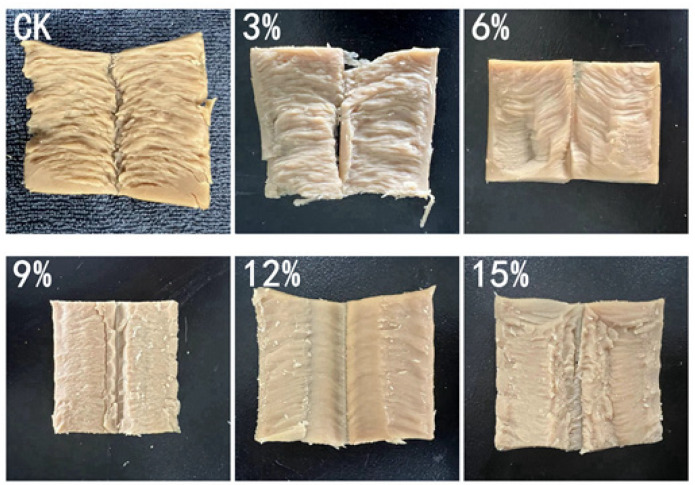
Profiles of extrudates with different amounts of SCP additions.

**Figure 3 foods-14-01134-f003:**
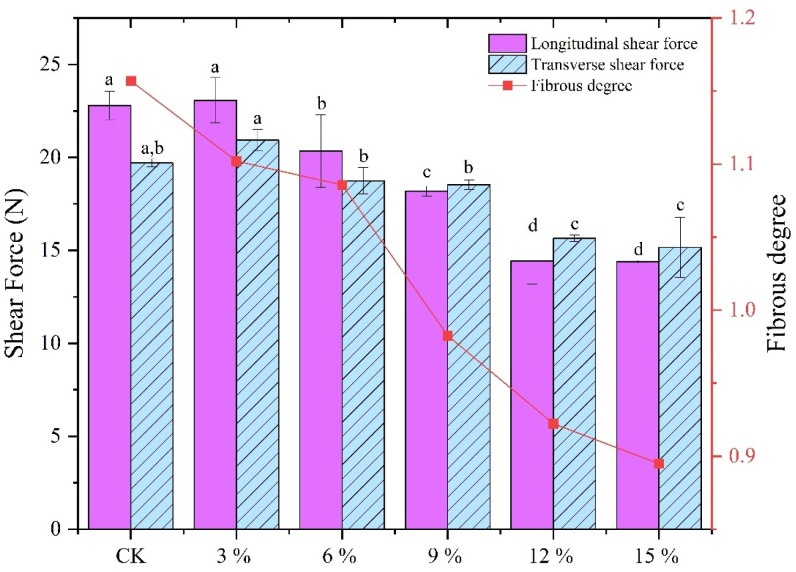
Effect of different SCP addition ratios on transverse and longitudinal shear force and fibrous degree of extrudates. Different lowercase letters superscripted indicate significant differences between extrusion property (*p* < 0.05).

**Figure 4 foods-14-01134-f004:**
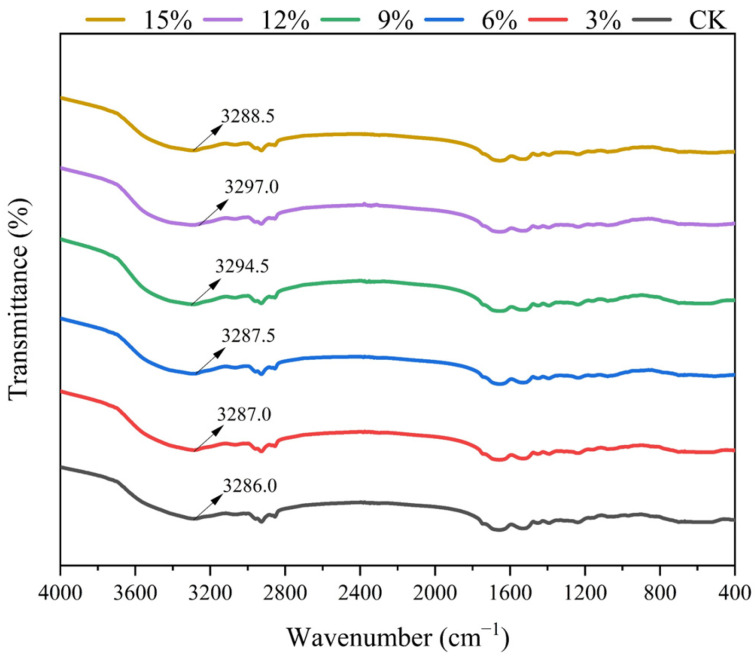
Fourier infrared spectra of extrudates with different proportions of SCP addition.

**Figure 5 foods-14-01134-f005:**
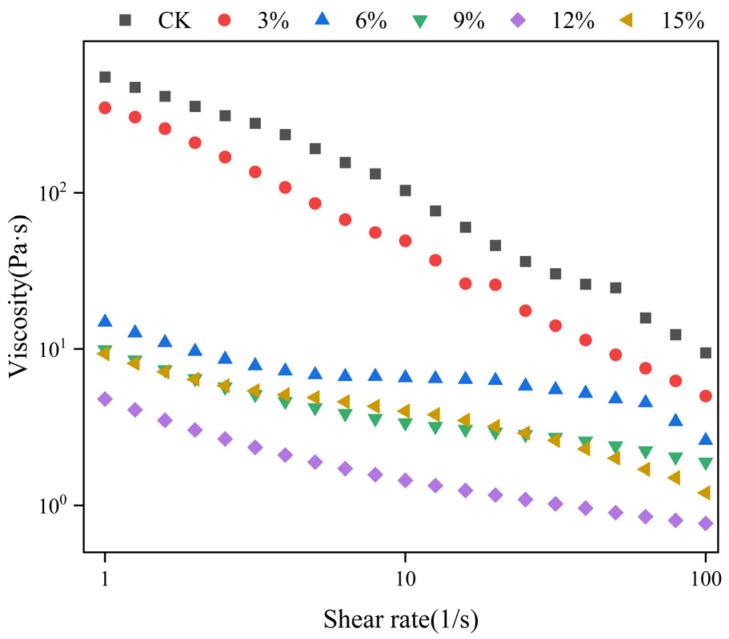
Effect of different SCP addition ratio on apparent viscosity of extrudate.

**Figure 6 foods-14-01134-f006:**
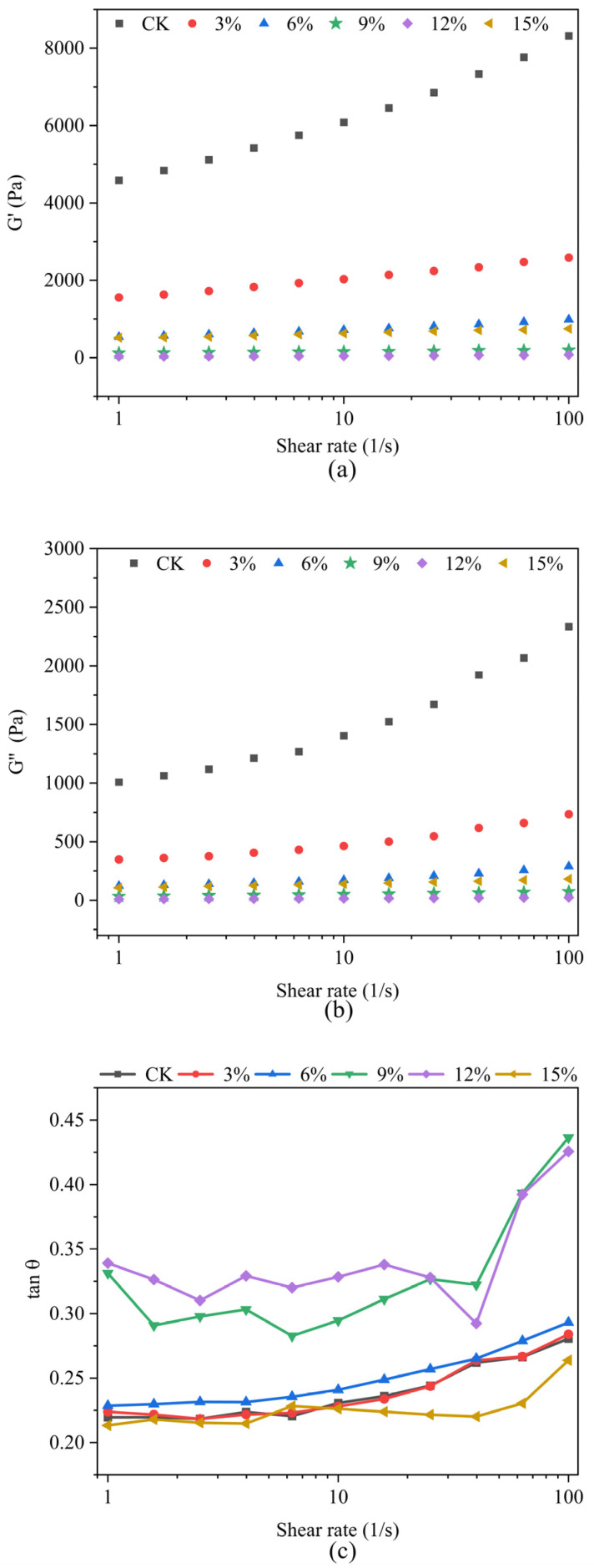
Storage modulus G′ and loss modulus G′′ in frequency scans of extrudates with different ratios of SCP addition (**a**,**b**); loss angle tangent (tanδ) of extrudates with different ratios of SCP addition (**c**).

**Figure 7 foods-14-01134-f007:**
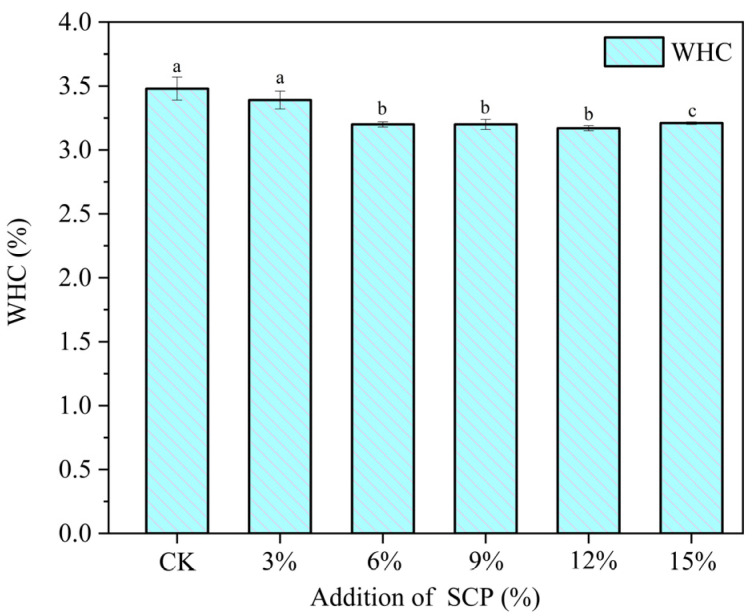
Effect of different percentages of SCP on the water-holding capacity of extrudates. Different lowercase letters superscripted indicate significant differences between extrusion property.

**Table 1 foods-14-01134-t001:** Effects of different SCP addition ratios on color parameters of extrudates.

Addition of SCP	*L**	*a**	*b**
%			
CK	64.26 ± 1.51 ^e^	1.44 ± 0.31 ^b^	17.02 ± 0.72 ^b^
3	65.60 ± 1.98 ^e^	0.01 ± 0.44 ^c^	17.06 ± 1.12 ^b,c^
6	68.30 ± 1.60 ^d^	0.11 ± 0.38 ^c^	17.71 ± 0.55 ^b,c^
9	71.31 ± 1.86 ^c^	−0.09 ± 0.49 ^c^	17.25 ± 1.14 ^b,c^
12	71.29 ± 1.02 ^c^	−0.07 ± 0.46 ^c^	17.20 ± 1.74 ^b,c^
15	71.61 ± 2.79 ^c^	−1.02 ± 0.87 ^d^	16.15 ± 0.58 ^c^
Raw PPI	80.24 ± 0.02 ^b^	2.45 ± 0.03 ^a^	19.54 ± 0.03 ^a^
Raw SCP	93.39 ± 0.01 ^a^	−2.87 ± 0.01 ^e^	8.86 ± 0.02 ^d^

Note: Different lowercase letters superscripted indicate significant differences between extrusion property (*p* < 0.05).

**Table 2 foods-14-01134-t002:** Effect of different SCP addition ratios on the texture characteristics of extruded materials.

Addition of SCP	Hardness	Adhesiveness	Resilience	Springiness	Gumminess	Chewiness
%	N	N·s	%	%		
CK	97.41 ± 1.13 ^a^	0.92 ± 0.23 ^a^	36.33 ± 0.44 ^a^	92.16 ± 1.20 ^a^	75.36 ± 1.19 ^a^	69.45 ± 1.91 ^a^
3	97.12 ± 3.95 ^a^	1.10 ± 0.34 ^a^	35.57 ± 1.06 ^a,b^	89.64 ± 2.77 ^a^	74.26 ± 2.10 ^a^	66.61 ± 3.91 ^a^
6	89.11 ± 0.21 ^b^	1.02 ± 0.12 ^a^	34.56 ± 0.27 ^b,c^	91.75 ± 0.60 ^a^	67.16 ± 0.71 ^b^	61.61 ± 0.48 ^b^
9	84.39 ± 1.39 ^c^	0.96 ± 0.10 ^a^	34.52 ± 0.77 ^b,c^	90.48 ± 1.07 ^a^	63.21 ± 1.09 ^c^	57.19 ± 1.48 ^c^
12	80.06 ± 0.90 ^d^	0.49 ± 0.14 ^b^	33.43 ± 0.58 ^c^	92.26 ± 1.30 ^a^	59.27 ± 1.01 ^d^	54.68 ± 1.06 ^c^
15	52.57 ± 0.97 ^e^	0.11 ± 0.02 ^c^	30.72 ± 0.82 ^d^	89.87 ± 0.43 ^a^	36.43 ± 0.98 ^e^	32.75 ± 0.86 ^d^

Note: Different lowercase letters superscripted indicate significant differences between extrusion property (*p* < 0.05).

**Table 3 foods-14-01134-t003:** Relative content of secondary structure of extrudates with different percentages of SCP addition.

Addition of SCP	Random Coil	β-Sheets	α-Helix	β-Turns	α-Helix/β-Sheets
%	%	%	%	%	
CK	2.41 ± 0.01 ^e^	58.92 ± 0.04 ^d^	24.42 ± 0.03 ^a^	14.26 ± 0.01 ^a^	0.4146 ± 0.0849 ^a^
3	4.35 ± 0.21 ^a^	63.90 ± 0.34 ^a^	21.19 ± 0.02 ^e^	10.57 ± 0.16 ^e^	0.3315 ± 0.1414 ^e^
6	3.33 ± 0.03 ^c^	60.66 ± 0.06 ^b^	23.03 ± 0.01 ^d^	12.99 ± 0.04 ^c^	0.3796 ± 0.0354 ^d^
9	3.52 ± 0.01 ^c^	60.74 ± 0.06 ^b^	23.08 ± 0.01 ^d^	12.61 ± 0.01 ^d^	0.3799 ± 0.0212 ^d^
12	2.95 ± 0.02 ^d^	59.40 ± 0.18 ^c^	23.85 ± 0.13 ^b^	13.81 ± 0.04 ^b^	0.4015 ± 0.3394 ^b^
15	3.57 ± 0.06 ^b^	60.56 ± 0.06 ^b^	23.31 ± 0.06 ^c^	12.56 ± 0.05 ^d^	0.3849 ± 0.1273 ^c^

Note: Different lowercase letters superscripted indicate significant differences between extrusion property (*p* < 0.05).

**Table 4 foods-14-01134-t004:** Power-law equation fitting results of static rheological properties of extrudates with different proportions of SCP addition.

Addition of SCP %	Power Law Equation: τ = K·γ ^n^ Fitting Results		
	K (Pa·s^n^)	*n*	R^2^
CK	602.06 ± 14.19 ^a^	0.25 ± 0.01 ^c^	0.98
3	307.73 ± 7.31 ^b^	0.20 ± 0.01 ^c^	0.96
6	20.45 ± 3.05 ^c^	0.64 ± 0.04 ^a,b^	0.95
9	7.13 ± 0.35 ^e^	0.72 ± 0.01 ^a^	0.99
12	3.33 ± 0.21 ^f^	0.67 ± 0.02 ^a,b^	0.99
15	14.89 ± 1.56 ^d^	0.57 ± 0.03 ^b^	0.97

Note: Different lowercase letters superscripted indicate significant differences between extrusion property (*p* < 0.05).

**Table 5 foods-14-01134-t005:** Amino acid content of extrudates with different proportions of SCP addition.

	Raw SCP	CK	3% SCP	6% SCP	9% SCP	12% SCP	15% SCP
Aspartic acid	5.26 ± 0.08 ^e^	8.66 ± 0.07 ^a^	8.48 ± 0.06 ^a,b^	8.34 ± 0.01 ^b,c^	8.24 ± 0.07 ^c,d^	8.14 ± 0.16 ^c,d^	8.07 ± 0.03 ^d^
Threonine	1.51 ± 0.02 ^e^	2.69 ± 0.02 ^a^	2.63 ± 0.02 ^a,b^	2.59 ± 0.01 ^b,c^	2.55 ± 0.02 ^c,d^	2.51 ± 0.05 ^d^	2.49 ± 0.02 ^d^
Serine	2.88 ± 0.04 ^d^	3.73 ± 0.06 ^a^	3.67 ± 0.02 ^a,b^	3.62 ± 0.01 ^b,c^	3.60 ± 0.01 ^b,c^	3.56 ± 0.07 ^b,c^	3.55 ± 0.01 ^c^
Glutamic acid	9.22 ± 0.09 ^d^	13.58 ± 0.16 ^a^	13.40 ± 0.11 ^a,b^	13.22 ± 0.01 ^b,c^	13.11 ± 0.08 ^b,c^	12.99 ± 0.25 ^c^	12.93 ± 0.07 ^c^
Proline	12.28 ± 0.05 ^a^	3.19 ± 0.03 ^g^	3.51 ± 0.04 ^f^	3.82 ± 0.02 ^e^	4.00 ± 0.01 ^d^	4.30 ± 0.07 ^c^	4.56 ± 0.04 ^b^
Glycine	24.23 ± 0.05 ^a^	2.99 ± 0.03 ^g^	3.58 ± 0.03 ^f^	4.23 ± 0.02 ^e^	4.71 ± 0.05 ^d^	5.28 ± 0.10 ^c^	5.84 ± 0.02 ^b^
Alanine	8.82 ± 0.02 ^a^	3.27 ± 0.03 ^g^	3.44 ± 0.03 ^f^	3.62 ± 0.01 ^e^	3.74 ± 0.03 ^d^	3.92 ± 0.08 ^c^	4.09 ± 0.02 ^b^
Cystine	0.04 ± 0.02 ^c^	0.63 ± 0.01 ^a^	0.63 ± 0.02 ^a^	0.61 ± 0.02 ^a^	0.56 ± 0.01 ^b^	0.57 ± 0.02 ^b^	0.53 ± 0.01 ^b^
Valine	2.08 ± 0.05 ^e^	3.87 ± 0.04 ^a^	3.80 ± 0.03 ^a,b^	3.73 ± 0.01 ^b,c^	3.68 ± 0.04 ^c,d^	3.64 ± 0.07 ^c,d^	3.58 ± 0.02 ^d^
Methionine	0.50 ± 0.03 ^e^	0.67 ± 0.01 ^a^	0.67 ± 0.01 ^a^	0.67 ± 0.01 ^a,b^	0.65 ± 0.01 ^c^	0.66 ± 0.01 ^b,c^	0.64 ± 0.01 ^d^
Isoleucine	1.29 ± 0.03 ^f^	3.68 ± 0.04 ^a^	3.59 ± 0.04 ^a,b^	3.50 ± 0.01 ^b,c^	3.43 ± 0.04 ^c,d^	3.38 ± 0.07 ^d^	3.27 ± 0.02 ^e^
Leucine	2.47 ± 0.05 ^e^	6.43 ± 0.06 ^a^	6.26 ± 0.06 ^b^	6.12 ± 0.03 ^b,c^	5.99 ± 0.05 ^c,d^	5.89 ± 0.11 ^d,e^	5.74 ± 0.04 ^e^
Tyrosine	0.20 ± 0.04 ^d^	2.74 ± 0.01 ^a^	2.69 ± 0.05 ^a^	2.55 ± 0.02 ^b^	2.49 ± 0.04 ^b^	2.39 ± 0.06 ^c^	2.36 ± 0.01 ^c^
Phenylalanine	1.2 ± 0.05 ^e^	4.02 ± 0.05 ^a^	3.95 ± 0.05 ^a^	3.81 ± 0.02 ^b^	3.72 ± 0.02 ^b,c^	3.65 ± 0.08 ^c,d^	3.57 ± 0.04 ^d^
Histidine	0.51 ± 0.04 ^f^	1.77 ± 0.01 ^a^	1.72 ± 0.02 ^a,b^	1.67 ± 0.01 ^b,c^	1.63 ± 0.01 ^c,d^	1.60 ± 0.03 ^d,e^	1.57 ± 0.01 ^e^
Lysine	3.60 ± 0.02 ^e^	5.56 ± 0.06 ^a^	5.46 ± 0.04 ^a,b^	5.38 ± 0.01 ^b,c^	5.29 ± 0.04 ^c,d^	5.25 ± 0.10 ^c,d^	5.18 ± 0.03 ^d^
Arginine	7.07 ± 0.08 ^a^	6.35 ± 0.06 ^b^	6.31 ± 0.05 ^b^	6.31 ± 0.01 ^b^	6.29 ± 0.01 ^b^	6.33 ± 0.12 ^b^	6.36 ± 0.05 ^b^
Tyrosine	0.03 ± 0.01 ^d^	0.57 ± 0.01 ^a^	0.55 ± 0.01 ^b^	0.54 ± 0.01 ^b^	0.49 ± 0.01 ^c^	0.50 ± 0.01 ^c^	0.50 ± 0.01 ^c^
∑	83.16 ± 0.38 ^a^	74.40 ± 0.12 ^g^	74.32 ± 0.15 ^f^	74.34 ± 0.36 ^e^	74.17 ± 0.62 ^d^	74.56 ± 0.76 ^c^	74.83 ± 0.86 ^b^

Note: Different lowercase letters superscripted indicate significant differences between extrusion property (*p* < 0.05).

**Table 6 foods-14-01134-t006:** Effect of different additive ratios of SCP on the thermal properties of extrudates.

Addition of SCP	*T_d_*	Δ*H*
%	°C	J/g
CK	94.63 ± 0.70 ^a^	137.30 ± 4.94 ^c^
3	93.40 ± 0.01 ^a^	232.15 ± 0.49 ^b^
6	92.01 ± 0.04 ^a,b^	237.22 ± 0.82 ^b^
9	89.63 ± 2.26 ^b^	232.05 ± 4.87 ^b^
12	92.24 ± 0.86 ^a,b^	252.60 ± 5.66 ^a^
15	93.49 ± 2.36 ^a^	238.35 ± 0.78 ^b^

Note: Different lowercase letters superscripted indicate significant differences between extrusion property (*p* < 0.05).

## Data Availability

The original contributions presented in the study are included in the article, further inquiries can be directed to the corresponding author.

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
