# Peer review of "Mechanism of Textural Reorganization in Silkworm Chrysalis and Pea Protein Extrusion: Structural Evolution and Quality Characteristic"

_foods, 2025, doi:10.3390/foods14071134_

Round 1

Reviewer 1 Report

Comments and Suggestions for Authors

This manuscript "Mechanism of Textural Reorganization in Silkworm Chrysalis and Pea Protein Extrusion: Structural Evolution and Quality Characteristic" investigates the mechanism of textural reorganization in silkworm chrysalis and pea protein extrudates, focusing on structural evolution and quality characteristics. While the research presents valuable data, discussing the results, particularly regarding textural properties and rheological behavior, requires significant expansion and contextualization. Comparisons with existing research and commercially available products are essential to establish the relevance and impact of the findings. Additionally, the manuscript must be strengthened by consistent use of international units and a more thorough integration of relevant literature on edible insect-based food formulations.

Specific Comments:

Lines 112-122: The content of these lines appears to be more appropriate for the methodology, results, or discussion sections. Consider relocating or removing these sentences to maintain a clear flow within the introduction.

Line 133: Please provide the rationale for the selection of the specific PPI:SCP ratios used in this study. Clarification of the selection criteria is necessary for the reader to understand the experimental design.

Line 285: The use of ascending and descending superscript letters to denote statistical differences in Table 2 is confusing. Consider adopting a consistent direction (e.g., all ascending or all descending) to improve clarity for the reader.

Line 287: The discussion of the textural characteristics of the extrudates is inadequate. Please expand this section by comparing your findings with relevant research and existing commercial products. This comparison is essential for demonstrating the significance of your results.

Line 310: The discussion of transverse and longitudinal shear force and the fibrous degree of the extrudates is insufficiently developed. Please provide a more in-depth analysis of these parameters and their implications.

Line 312: Please provide citations and supporting arguments for the statement made in this line.

Line 403: While the analysis of small amplitude oscillation shear stress measurements is well-contrasted within this study, a more comprehensive comparison with existing literature is needed to contextualize the findings.

Edible Insect-Based Food Formulations: The discussion should be expanded to include a review of current edible insect-based food formulations produced using advanced technologies, such as humid extrusion. This will strengthen the justification for your novel food formulation and highlight its advantages over existing products.

International Units: Please ensure consistent use of international units throughout the manuscript (e.g., 'mL' for milliliters, 'h' for hours). Inconsistencies like 'ml' and 'mL', or 'h' and 'hours' detract from the professional presentation of the work.

Reviewer 2 Report

Comments and Suggestions for Authors

The manuscript "Mechanism of Textural Reorganization in Silkworm Chrysalis and Pea Protein Extrusion" explores how mixing silkworm chrysalis protein (SCP) with pea protein isolate (PPI) during high-moisture extrusion affects their structure and functionality. This is an exciting and innovative study, particularly given the rising interest in alternative protein sources. While the manuscript is well-organized and presents valuable insights, some areas could be improved to make it clearer, more in-depth, and scientifically robust. Below are my suggestions.

  1. Why Choose SCP?

    • The study introduces SCP as a promising protein but does not fully explain why it was selected over other insect-based proteins. Including a comparison with other options, highlighting its unique benefits in terms of nutrition, processing, and sustainability, would add value.

  2. How Do SCP and PPI Interact?

    • The manuscript shows that extrusion alters protein structure but does not dive deeply into the mechanisms behind these changes. Adding more details on how SCP and PPI interact at a molecular level—especially regarding bonding and structural reorganization—would improve the discussion.

  3. Data and Reproducibility:

    • The manuscript presents data effectively, but additional details on sample sizes, statistical power, and experiment reproducibility would improve credibility. Clarifying how many replicates were performed and discussing potential variability in the results would be beneficial.

  4. Market Relevance and Consumer Acceptance:

    • The paper suggests that SCP enhances market acceptance but does not provide data or references to support this claim. Including sensory analysis results or discussing consumer perception studies would make this argument more convincing.

Minor Issues and Suggested Improvements:

  1. Abstract and Introduction:

    • The abstract should be more concise and structured, emphasizing key findings more clearly.

    • The introduction should provide a broader background on SCP’s role in food technology beyond being an alternative protein.

  2. Figures and Tables:

    • Some figures (such as SEM images and rheological data) need clearer labels and higher resolution for better interpretation.

    • Tables should specify statistical significance where applicable.

  3. Technical Consistency:

    • Some terms (e.g., "entanglement" vs. "aggregation") are used inconsistently. Standardizing terminology would make the text more readable.

  4. Updating References:

    • Some cited studies are outdated. Adding more recent references would give the manuscript a stronger foundation.

  5. Process Optimization:

    • While the manuscript discusses extrusion parameters, it does not suggest ways to optimize the process. A section on improving temperature control, screw speed, or mixing ratios would add practical value.

This study introduces a novel approach to hybrid protein extrusion, which has strong potential for sustainable food development. By refining the discussion on SCP’s role, explaining the molecular interactions in more detail, and strengthening the statistical analysis, the manuscript could significantly improve its clarity and impact.

Comments on the Quality of English Language

The quality of English in the article is generally clear and understandable, but there are some areas where it could be improved for fluency, conciseness, and technical clarity. Here are some key observations:

  • Grammar and Syntax Issues – Some sentences contain minor grammatical errors or awkward phrasing. For example:

    • "These results systematically elucidated the microscopic scale mechanism changes of SCP and PPI during protein mixing and restructuring under a high moisture extrusion environment."
      → Could be rewritten for better clarity:
      "These results systematically elucidate the microscopic mechanisms of SCP and PPI restructuring during high-moisture extrusion."

    • "The addition of SCP facilitates the expansion of the mixed protein chain during the mixing and cooking phases of the extrusion process."
      "The addition of SCP promotes the expansion of the mixed protein chain during extrusion." (More concise)

  • Redundant or Wordy Sentences – Many sentences could be more concise without losing meaning. For example:

    • "This study employed SCP and PPI with varying mixing ratios for HME, with a particular focus on the conformational alterations of animal and plant proteins during microstructure reorganization under conditions of elevated temperature and pressure."
      "This study examined the conformational changes of SCP and PPI under high-moisture extrusion at elevated temperatures and pressures." (More direct)
  • Use of Passive Voice – The paper frequently uses passive voice, which can make sentences feel less engaging. For example:

    • "The extrusion process was conducted using a twin-screw extruder."
      "We conducted the extrusion process using a twin-screw extruder." (Active voice is often preferable for clarity)
  • Inconsistent Use of Articles (a/an/the) – There are occasional missing or misplaced articles. Example:

    • "The study demonstrated that modifying the SCP ratio in PPI under high-moisture extrusion has a marked impact on the structural and functional characteristics of the extrudates."
      "The study demonstrates that modifying the SCP ratio in PPI under high-moisture extrusion significantly impacts the structural and functional characteristics of the extrudates."

Reviewer 3 Report

Comments and Suggestions for Authors

The manuscript is interesting, but minor corrections need to be made. For example, in the methodology part, explain why it is the only insect used and the nutritional importance of plant-based proteins and their digestibility. The conclusions mention why a decrease in texture, water retention capacity, and microstructure is better if the opposite is usually sought. 
A file with comments on the manuscript is attached.
